# Microsatellite Instability: Diagnosis, Heterogeneity, Discordance, and Clinical Impact in Colorectal Cancer

**DOI:** 10.3390/cancers11101567

**Published:** 2019-10-15

**Authors:** Camille Evrard, Gaëlle Tachon, Violaine Randrian, Lucie Karayan-Tapon, David Tougeron

**Affiliations:** 1Department of Medical Oncology, Poitiers University Hospital, 86021 Poitiers, France; camille.evrard@chu-poitiers.fr; 2Department of Cancer Biology, Poitiers University Hospital, 86021 Poitiers, France; gaelle.tachon@chu-poitiers.fr (G.T.); lucie.karayan-tapon@chu-poitiers.fr (L.K.-T.); 3Faculty of Medicine, University of Poitiers, 86000 Poitiers, France; violaine.randrian@orange.fr; 4Laboratory of Experimental and Clinical Neuroscience, Institut national de la santé et de la recherche médicale (INSERM) 1084, F-86073 Poitiers, France; 5Department of Gastroenterology, Poitiers University Hospital, 86021 Poitiers, France

**Keywords:** microsatellite instability, colorectal cancer, immune checkpoints, deficient mismatch repair

## Abstract

Tumor DNA mismatch repair (MMR) deficiency testing is important to the identification of Lynch syndrome and decision making regarding adjuvant chemotherapy in stage II colorectal cancer (CRC) and has become an indispensable test in metastatic tumors due to the high efficacy of immune checkpoint inhibitor (ICI) in deficient MMR (dMMR) tumors. CRCs greatly benefit from this testing as approximately 15% of them are dMMR but only 3% to 5% are at a metastatic stage. MMR status can be determined by two different methods, microsatellite instability (MSI) testing on tumor DNA, and immunohistochemistry of the MMR proteins on tumor tissue. Recent studies have reported a rate of 3% to 10% of discordance between these two tests. Moreover, some reports suggest possible intra- and inter-tumoral heterogeneity of MMR and MSI status. These issues are important to know and to clarify in order to define therapeutic strategy in CRC. This review aims to detail the standard techniques used for the determination of MMR and MSI status, along with their advantages and limits. We review the discordances that may arise between these two tests, tumor heterogeneity of MMR and MSI status, and possible explanations. We also discuss the strategies designed to distinguish sporadic versus germline dMMR/MSI CRC. Finally, we present new and accurate methods aimed at determining MMR/MSI status.

## 1. Introduction

There are three major mechanisms in colorectal (CRC) carcinogenesis. The most common is chromosomal instability (CIN) in 75% of CRCs, the second is an epigenetic modification of DNA methylation, also called CpG island methylator phenotype (CIMP), in 20% of CRCs, and the third is microsatellite instability (MSI) or deficiency of DNA mismatch repair system (MMR) which occurs in approximately 15% of CRCs (Figure 1) [1]. It is worth noting that there is frequent overlap between CIMP and dMMR/MSI phenotype. CRC carcinogenesis is somewhat more complex, and while rare overlaps between CIN and CIMP or CIN and MSI phenotypes exist, they will not be developed here [2].

Microsatellites (also called “short tandem repeats”) correspond to DNA sequences distributed throughout the genome (coding or non-coding sequences) with a repetitive structure, i.e., repetition, a variable number of times, of a single nucleotide or di-, tri-, or tetra-nucleotides. These repetitive structures are particularly prone to replication errors in the case of deficiency of the MMR system (dMMR status). An accumulation of errors in the sequence of these microsatellites, called microsatellite instability, highlights malfunction of the MMR system. Cancers with such phenotypes are said to be MSI or by extension, MMR-deficient (dMMR), which is the opposite of microsatellite stability (MSS), also known as MMR-proficient (pMMR).

The MMR system consists of four major proteins called MLH1, MSH2, MSH6, and PMS2. These proteins identify and correct DNA mismatches caused by DNA polymerase during replication, which occurs especially in microsatellites. These proteins work two by two, MLH1 with PMS2 and MSH2 with MSH6, and form the MutLα and MutSa complex, respectively. MutSa recognizes single base pair mismatch, creates a sliding clamp around DNA, and then binds the second complex, MutLα [3]. This combination interacts with many enzymes, including the DNA polymerase, to perform excision of the single mismatch and resynthesize the DNA strand (Figure 2). MutSa can also recognize other error patterns such as insertion and deletion loops and then preferably bind with MutL homologs, such as MutLβ (MLH1-PMS1) and MutLγ (MLH1-MLH3), thereby achieving error excision [4,5]. Loss of function of one of the four proteins (MLH1, PMS2, MSH2, or MSH6) leads to inactivation of the MMR system, resulting in a loss of fidelity of the replication and an accumulation of mutations. This ultra-mutated profile causes dMMR/MSI cancers, including dMMR/MSI CRC.

The dMMR/MSI phenotype can be acquired in sporadic CRC (75%) or constitutively in Lynch syndrome, due to germline mutation of one of the MMR genes (25%) [6]. Sporadic cases are due in most cases to a loss of expression of MLH1 caused by hypermethylation of the *MLH1* gene promoter. Indeed, dMMR/MSI detection is of major help to identify Lynch syndrome according to the revised Bethesda criteria (Table 1). Other clinical impacts of dMMR/MSI determination enter into decision making regarding adjuvant chemotherapy in stage II CRC and the use of immune checkpoint inhibitor (ICI) in chemoresistant metastatic dMMR/MSI tumors. About 20% of stage II and III CRCs present a dMMR/MSI phenotype and are associated with better prognosis than pMMR/MSS tumors [5]. Moreover, stage II dMMR/MSI CRCs do not benefit from adjuvant fluoropyrimidine chemotherapy [7,8]. Consequently, in view of a good prognosis for stage II dMMR/MSI CRCs and chemoresistance to fluoropyrimidine, adjuvant chemotherapy is not recommended. Nevertheless, for high-risk stage II dMMR/MSI CRCs with very poor prognosis criteria, such as T4 stage and vascular emboli, oxaliplatin-based adjuvant chemotherapy should be discussed case by case. In metastatic CRCs (mCRC), dMMR/MSI represents only around 3% to 5% of mCRCs and has been associated with poor prognosis and chemoresistance to standard treatment [9,10]. Nevertheless, recent series have reported prolonged overall survival in dMMR/MSI mCRC and a trend toward better outcomes of anti-vascular endothelial growth factor (anti-VEGF) as compared with anti-epidermal growth factor receptor (anti-EGFR), but no difference according to chemotherapy regimen, i.e., irinotecan-based chemotherapy as compared with oxaliplatin-based chemotherapy has been reported [11]. Finally, recent nonrandomized trials suggest high efficacy of immune checkpoint inhibitor (ICI) in chemoresistant dMMR/MSI metastatic tumors due to the high tumor mutational burden in these tumors, while the other predictor of response to ICI is the IHC labeling of the PD-L1 protein (programmed death-ligand 1) [12,13].

To conclude, dMMR/MSI testing in CRC is indicated primarily in the following three circumstances: in stage II CRCs to define indications and modalities of adjuvant chemotherapy, in stage IV to treat with ICI, and for screening of Lynch syndrome based on the revised Bethesda criteria. However, in some centers, dMMR/MSI testing is performed in all CRCs given its interest in multiple circumstances.

## 2. Mismatch Repair System and Microsatellite Instability Testing

The status of dMMR and MSI can be determined by two different methods, molecular MSI testing based on DNA extracted from tumor tissue and immunohistochemistry (IHC) of the MMR proteins based on sections of formalin-fixed paraffin-embedded (FFPE) tumor block. Commonly, a tumor is called dMMR if it presents nuclear loss of expression of at least one of the MMR proteins (MLH1, PMS2, MSH2, or MSH6), in contrast to pMMR tumor. A tumor is called MSI (or MSI-high, MSI-H) if it presents instability (variation in the microsatellite length) of at least 40% of a panel of microsatellites tested on tumor DNA, in contrast to MSS tumor.

### 2.1. Microsatellite Instability Testing

Microsatellite instability testing is carried out on tumor DNA extracted from frozen or FFPE tumor tissue. A specific area with high tumor cellularity (>20%) has to be selected by an experienced pathologist to avoid false negative results [15]. Microsatellite instability is assessed by analyzing microsatellite loci, either mononucleotide repeats or a combination of dinucleotide and mononucleotide repeats as recommended [16].

The first reference panel, referred to as the Bethesda panel, consists of two mononucleotide loci (big adenine tract *BAT*-*25* and *BAT-26*) and three dinucleotide loci (*D2S123*, *D5S346*, and *D17S250*) [12]. *BAT-25* is found in intron 16 of the *c-KIT* gene and *BAT-26* in the intron 5 of the *MSH2* gene. *D2S123* (2p16) is telomeric to mismatch repair genes *MSH-2* and *MSH-6*, *D5S346* (5q21–22) is close to the locus for the *adenomatosis polyposis coli* (*APC*) tumor suppressor gene, and *D17S250* (17q11.2–12) is close to the locus for the tumor suppressor gene *BRCA-1* [17]. As dinucleotide repeats are less sensitive than mononucleotide repeats for MSI detection, a comparison with non-tumor matching DNA is always required (DNA extracted from non-tumor tissue from a surgical specimen). When using the Bethesda panel, tumors with instability of two or more of these loci are considered as MSI (i.e., MSI-high), and cancers with no instability at any of the five loci are considered MSS. Cancers with only one out of five unstable loci are interpreted as MSI-low, although it is unclear whether MSI-low represents a biologically distinct category or whether it simply reflects the inherent limitations of faithfully replicating these repetitive sequences. For the time being, these tumors are considered as MSS.

The second panel is the pentaplex panel, which consists of the following five consensus mononucleotide repeats: *BAT-25*, *BAT-26*, *NR-21* located in the untranslated 5’ region of the *SLC7AB* gene, *NR-24* located in the untranslated 5’ region of the *ZNF-2* gene, and *NR-27* (*MONO-27*) located in the untranslated 5’ region of the *IAP-1* gene. The quasi-monomorphic nature of these microsatellites facilitates their analysis; they have several repetitions and their size is highly homogeneous in the Caucasian population [18]. Using the pentaplex panel, a tumor will be called MSI if at least three markers out of five are unstable (Figure 3B). A tumor with no instability, or one unstable marker, is classified MSS (Figure 3A). In very rare cases, where two markers are unstable, a healthy tissue analysis is carried out to confirm or deny instability by comparing healthy and tumor tissue. This panel has become the new standard, in most international recommendations for MSI testing; contrary to the Bethesda panel, it does not require comparison with non-tumor tissue for MSI determination in CRC [19].

Both MSI techniques are based on the simultaneous amplification of five markers in a multiplex PCR. Amplification products then migrate on capillary electrophoresis, which enables marker distribution depending on their sizes. Taq polymerase is not faithful enough to reproduce the exact number of nucleotide repetitions, and therefore, at the expected size of a marker, the electrophoresis profile shows a multiple peak pattern. The reference size of the marker per sample is given by the main, higher peak and needs to be compared to its expected size in the general population [16]. If the size of the marker significantly differs from its expected size, it signifies instability. For non-colorectal tumor tissue, systematic double testing of tumor tissue and non-tumor tissue is performed for MSI testing. When compared with non-tumor tissue, the threshold to define MSI is lowered to two unstable markers out of five [20].

### 2.2. MMR Protein Testing

IHC is a technique detecting protein expression directly from tissue samples. As MMR proteins (MLH1, MSH2, MSH6, and PMS2) are in the nucleus, only nuclear staining is taken into account in MMR protein detection. MMR protein loss is defined by the absence of IHC staining in the nucleus of tumor cells, whereas normal cells remain stained, ensuring the technical validity of the experiment. Indeed, nuclear staining of non-cancerous stromal cells is considered a good internal positive control, even in Lynch syndrome, insofar as only one allele is mutated outside the tumor area (germline mutation of one allele of one MMR gene). A cell develops a DNA repair defect only when its second copy of the gene also becomes non-functional (Knudson’s two-hit hypothesis) as a result of a random mutation (somatic mutation of the second allele of the same MMR gene). A loss of MMR protein expression means genomic alteration (loss of heterozygosity, mutations or epigenetic modifications) of the corresponding gene. A tumor is considered dMMR if one of the four MMR proteins is lost (absence of nuclear staining) (Figure 4).

In the MMR system, protein functions are achieved by heterodimers, MLH1 being the PMS2 partner and MSH2 being the MSH6 partner. In their monomeric form, MMR proteins are degraded. Consequently, the loss of one MMR protein is usually accompanied by the loss of its partner. For instance, loss of MLH1 or MSH2 proteins will leave PMS2 and MSH6, respectively, in their monomeric forms, and they will be rapidly degraded. Consequently, loss of MLH1 expression is associated with PMS2 loss and loss of MSH2 expression is associated with MSH6 loss. However, the contrary is not true insofar as MLH1 and MSH2 proteins, in their monomeric forms, can interact with other proteins of the MMR system, i.e., MSH3, and thereby avoid degradation [21]. Indeed, isolated loss of PMS2 expression (≈5% to 10% of dMMR CRC [22]) or isolated loss of MSH6 expression (≈5% to 15% of dMMR CRC [23]) is not rare. In view of both economy and time saving, some pathologists have analyzed MMR expression of two proteins (MLH1 and MSH2) instead of four but have not been able to detect isolated loss of PMS2 or MSH6 expression. Indeed, most guidelines recommend performing IHC of all four MMR proteins to avoid misinterpretation [24].

### 2.3. Comparison of Molecular MSI Testing and MMR Proteins Immunohistochemistry

The molecular approach (MSI) presents the advantage of studying MMR system dysfunction and is not limited to protein expression. Indeed, some point mutations allow MMR protein expression (normal MMR proteins IHC), but without retaining the MMR function (MSI). However, MSI testing takes somewhat longer and costs a little more than the IHC technique and does not provide information as to which MMR gene is deficient.

While both MSI panels, the Bethesda and the pentaplex, use the two microsatellite markers *BAT-25* and *BAT-26*, polymorphisms have been reported for both, especially in African ethnicities [25]. To detect these polymorphisms and to avoid counting them as unstable markers, it is commonly admitted that an isolated unstable marker must be validated in comparison with the microsatellite profile of its matched non-tumor tissue, even for CRC using the pentaplex panel. In this case, MSI exploration will take longer. Finally, MSI interpretation requires trained molecular biologists, particularly to avoid misinterpretation of *BAT-25* or *BAT-26* microsatellites [25,26].

MMR protein IHC is easier and less expensive. Moreover, an isolated loss of MSH6 is not always responsible for MMR system deficiency, and hence not always detected by MSI technique [27,28]. Indeed, there exists functional redundancy of MMR proteins (PMS2 and PMS1, MSH6 and MSH3) with MMR protein expression loss at IHC loss, while MSS status is retained due to partial activity of the MMR system [22]. Moreover, MMR protein IHC can be difficult or misleading as it depends on staining processes, which are not standardized from one laboratory to another (antigen demasking, optimal antibody dilution, incubation time, etc.). Indeed, MMR protein IHC requires experienced pathologists [29]. The technique can give false positive results as a non-functional MMR protein can remain expressed in tumor tissue and detected by IHC even though the tumor is MSI. Indeed, although one third of *MLH1* mutations are missense mutations coding for non-functional proteins, MLH1 expression remains detected by IHC [28]. Heterogeneous loss of MMR protein expression has also been observed for MSH6 due to *MSH6* somatic mutations [30].

With regard to sensitivity and specificity of IHC, they vary from 81% to 100% and from 80% to 92%, respectively [31]. MSI analysis sensitivity ranges from 67% to 100% and specificity from 61% to 92% using the Bethesda panel. When using the pentaplex panel, sensitivity is better (89% to 100%) (no control with non-tumoral tissue) as is specificity (79% to 100%) [32,33].

## 3. Challenges for Determination of the dMMR/MSI Mechanism

### 3.1. How to Classify Sporadic versus Germline dMMR/MSI Colorectal Cancer?

MMR deficiency can be acquired in sporadic dMMR/MSI tumors or constitutively in Lynch syndrome-related dMMR/MSI tumors. About 75% of dMMR/MSI CRCs are sporadic cases. In most cases, observed loss of MLH1 expression is caused by hypermethylation of its promoter, acquired during tumorigenesis. This epigenetic modification is developed in the context of senescence with global hypermethylation of DNA, mainly at CpG islands [34]. CpG islands are regions of DNA that contain numerous Guanine Cytosine dinucleotides bound together by a phosphodiester and frequently found in gene promoters. Usually, when the cytosine is methylated in the promoter region, it causes inhibition of transcription of the gene. CRC with global DNA hypermethylation is called “hypermethylator” or CpG island methylator phenotype (CIMP). CIMP tumors are frequently associated with *BRAF* p.V600 mutation (from 77% to 95%) but their interconnections remain unclear [35,36]. Therefore, dMMR/MSI CRCs with MLH1 loss and *BRAF* mutation are considered as sporadic dMMR/MSI CRCs. In the case of dMMR/MSI CRC with MLH1 loss and *BRAF* wild type status, *MLH1* promoter methylation is determined and a *MLH1* promoter hypermethylation signs the sporadic trait of the tumor.

The second group of dMMR/MSI CRCs is primarily related to Lynch syndrome (LS), i.e., with detectable monoallelic germline mutation in one of the four MMR genes. The LS diagnosis is a confirmed mutation of one of the MMR genes. The percentage of mutations identified in the case of suspicion of LS is very variable according to the criteria selected, i.e., more than 90% in the case of loss of MSH2 or MSH6, about 70% if the Amsterdam II criteria are met, about 40% in case of loss of MLH1, and about 30% if the revised Bethesda criteria are met [37,38,39]. Most of the germline mutations in LS occur in *MLH1* or *MSH2* genes (90%) (Table 2). *MSH6* and *PMS2* genes are less frequently affected (about 10% of cases). In addition, the pathogenicity of *PMS2* mutations is more difficult to establish because of pseudogene interferences [40]. Pseudogenes are usually characterized by a combination of homology to a known gene and loss of some functionality. Indeed, a *PMS2* non-expressed pseudogene exists and is highly homologous in both intronic and exonic sequences to the *PMS2* gene, hence polymorphisms in this pseudogene can be mistaken for mutations in the *PMS2* gene.

According to the model described by Knudson et al. for tumor suppressor genes such as MMR genes, one allele of the MMR gene has a germline mutation (first hit) and the second allele of the same MMR gene is inactivated at the somatic level (second hit) by various alterations (somatic mutation, genomic rearrangement, promoter hypermethylation, and loss of heterozygosity), which induce deficiency of the MMR system [41]. International guidelines have defined algorithms to efficiently identify LS patients among patients with sporadic dMMR/MSI tumors (Figure 5) [42].

### 3.2. MLH1 Promoter Hypermethylation

In dMMR/MSI CRC with loss of expression of MLH1, identification of the *MLH1* promoter hypermethylation and *BRAF* p.V600 mutation certifies the sporadic origin [24]. Conversely, if these alterations are not found, the patient should be referred to oncogenetics for MMR germline testing. Most teams initially limit their exploration to *BRAF* status due to cost and the time required for difficult *MLH1* promoter hypermethylation testing. In the event of dMMR/MSI CRC with loss of expression of MLH1 and *BRAF* wild-type status *MLH1* promoter hypermethylation testing is performed secondarily. Nevertheless, about 1% to 2% of LS cases (dMMR/MSI CRC with germline mutation) carry *BRAF* mutation [43]. These cases are rare but, at the individual level, the mis-screening of LS can have major consequences for the patient and his relatives. Indeed, dMMR/MSI *BRAF*-mutated CRC should be tested for *MLH1* promoter hypermethylation in the case of a high suspicion of LS to confirm, before germline MMR testing, that it is not a sporadic case.

Determination of somatic versus germline mechanism of dMMR/MSI CRC using methylation of the promoter of *MLH1* gene can sometimes be incorrect. While hypermethylation is mostly a somatic event, several cases of CRC with constitutional epimutations of the *MLH1* gene have been reported [44,45,46]. Constitutional *MLH1* epimutations cause severe LS phenotype, including young age of cancer onset and multiple primary tumors [45]. Usually, constitutional *MLH1* epimutation arises de novo with no or non-Mendelian inheritance and the second hit has a genetic, not an epigenetic basis (somatic mutation or loss of heterozygosity) [46]. Patients with dMMR/MSI CRC with *MLH1* promoter hypermethylation should be screened for constitutional *MLH1* epimutations in the case of early onset CRC (before 50 years) or with multiple tumors before 60 years [47].

### 3.3. Challenge in Determination of Lynch Syndrome

In approximately 60% to 70% of dMMR/MSI CRCs with suspected LS according to the revised Bethesda criteria, no mutation is identified (or variants of unknown significance) [48,49]. These tumors are called Lynch-like syndrome (LLS) and are a challenge for predisposition management of the patient and his family [48]. Indeed, the risk of colorectal cancer is higher in these families than in the general population but lower than in LS families [50]. LLS can be due to biallelic somatic inactivation of one MMR gene (or an allelic somatic mutation associated with loss of heterozygosity of the other allele), which means that the progeny of the patient has a risk of CRC equal to the general population. Indeed, upon excluding dMMR/MSI CRC patients with *MLH1* promoter hypermethylation and germline mutations, biallelic somatic inactivation is responsible for approximately 50% of the remaining dMMR/MSI CRCs [51]. Despite the evident interest of somatic exploration to avoid expensive and stressful screening protocol for a family with LLS, these tests are not routinely performed. Moreover, some of them remain unexplained and may be due to genetic alterations that have yet to be identified.

Another issue is the 3′ deletion of the *EPCAM* gene (epithelial cell adhesion molecule), also called *TACSTD1* (tumor-associated calcium signal transducer 1), which leads to a hypermethylator profile of its neighbor gene, *MSH2* [37]. The latter will not be expressed and a loss of MSH2/MSH6 protein will be detected by IHC. As no mutation in MMR genes will be detected by germline testing, the tumor can wrongly be classified as LLS with low cancer risk, whereas it is known that carriers of an *EPCAM* deletion have a cumulative risk of CRC similar to carriers of *MSH2* mutation [37]. Nevertheless, in most laboratories, germline *EPCAM* deletions are now analyzed primarily or secondarily in the absence of MMR mutation. The incidence of *EPCAM* deletions has appeared to vary between populations and may explain at least 1% to 3% of LS [52].

All in all, dMMR/MSI CRCs without evident sporadic mechanisms (i.e., *MLH1* promoter hypermethylation) and without germline mutation of *MLH1, PMS2, MSH2*, *MSH6*, deletion of *EPCAM* or germline *MLH1* promoter hypermethylation are considered unclassified as regards the molecular mechanism underlying the MMR deficiency, LS versus sporadic cases. This situation accounts for at least 30% of dMMR/MSI CRC patients with suspected LS [50,53]. These tumors are mostly 1-loss of MSH6 protein expression without *MSH6* germline mutation, 2-loss of MSH2 without *MSH2* germline mutation or *EPCAM* deletion, 3-loss of PMS2 with no loss of MLH1 and no *PMS2* germline mutation, and 4-loss of MLH1 protein expression with no *BRAF* mutation and no *MLH1* promoter hypermethylation and no *MLH1* germline mutation. These patients and their first-degree relatives must be considered as LS and monitored as such.

## 4. Discordance Between MMR Immunohistochemistry and DNA Microsatellites Testing

On the basis of the literature, discordances between IHC of MMR proteins and MSI molecular testing results range from 1% to 10% (Table 3).

Studies evaluating discordances between molecular MSI and IHC tests in CRC are limited. Moreover, it is difficult to compare these studies with each other because the molecular panels used for MSI testing and the antibodies used for IHC are not the same. It is worth noting that the most frequently observed discordance was loss of MSH6 expression with MSS status [23]. Indeed, dMMR/MSS CRC with an isolated loss of MSH6 could be due to the partial redundancy of MSH6 and MSH3 protein function. When the MSH6 protein is impaired, the MSH2/MSH3 heterodimer continues to operate and DNA mismatch errors are partially corrected [26].

Unpublished data from our retrospective series of 1085 CRC patients showed 2.3% of discordances using IHC of the four MMR proteins and the pentaplex panel. Among the 25 discordant cases (2.3%), reviewing by expert biologists and pathologists enabled reclassification of seven cases, mostly because of misinterpretation of IHC due to poor quality of the staining or few tumor cells in the biopsy. The remaining 18 discordant cases (1.7%) were mainly dMMR/MSS tumors (*n* = 15/18) with isolated loss of MSH6 (*n* = 6). Indeed, pathologist and biologist expertise is crucial for accurate determination of dMMR and MSI status, as also described by Jaffrelot et al. [61]. They studied 2528 patients with different types of cancers with a discordance rate of 1.1%, using pentaplex panel for molecular biology and four proteins for IHC. Cohen et al. studied fewer patients (*n* = 92) but reported a higher discordance rate (9.8%) with the same two methods (pentaplex and four proteins) [60].

To avoid these discordances due to technical issues, it is necessary to follow some rules when performing the tests, mainly to use tumor samples with good quality, more than 20% of tumor cells and before any treatment, if possible (Table 4). Moreover, in cases of discordance, both tests must be repeated so as to detect errors or tumor heterogeneity.

Finally, 1% to 2% of discordances between MMR protein IHC and MSI molecular testing by pentaplex remain unexplained. In these cases, germline mutation testing of the MMR genes must be performed if LS is suspected. A diagnostic of LS implies a lifelong cancer screening protocol for the patient and their relatives. Moreover, a recent report suggests that half of the diagnoses of patients with mCRC and primary resistance to ICI are due to misinterpretation of the MMR IHC or MSI tests. Indeed, to avoid false positive of dMMR or MSI tests, both tests are recommended before ICI treatment, especially in clinical trials (Figure 6). Indeed, most trials with ICI now consider these discordances during patient inclusion. Whereas in previous studies, either dMMR or MSI status was required for patient inclusion [65,66], at present, both tests have to be performed with no discordance (dMMR and MSI) [67,68]. Patients with discordant tests are not eligible for these ongoing trials (pMMR/MSI or dMMR/MSS).

Research is ongoing to improve detection of dMMR/MSI cases and to avoid discordant cases, with particular interest in the use of other microsatellite markers such as HSP110 [69] or an increased number of microsatellites analyzed using next-generation sequencing (NGS) [70] (see paragraph “perspective of MSI/MMR testing”).

## 5. Focus on Tumor Heterogeneity

Although microsatellite instability is considered as an early event in CRC carcinogenesis, several recent studies have reported that microsatellite instability is not always a homogeneous event throughout the tumor in sporadic CRC [64,71]. Intra-tumoral heterogeneity is defined as the emergence of tumor subclones with different genotypes in the same tumor mass and inter-tumoral heterogeneity consists of the presence of at least two different tumor subclones on different tumor sites. Most studies in mCRC have reported good concordance in mutational profiles of major signaling pathways, such as *KRAS, TP53, APC, PIK3CA, BRAF,* and *NRAS*, between the primary tumor and its metastases, superior to 95% [72,73]. However, some recent studies using highly sensitive techniques or microdissection have identified tumor heterogeneity in CRC. Intra-tumoral and inter-tumoral heterogeneity of *KRAS* mutations is now well-known in mCRC and correlates with resistance or reduced efficacy of anti-EGFR therapies [74]. For instance, the Jeantet et al. study identified a high rate of *RAS* mutation heterogeneity in mCRCs with 33% of intra-tumoral heterogeneity and 36% of inter-tumoral heterogeneity [75]. Most articles about tumor heterogeneity have focused on common mutations such as *KRAS, TP53,* and *PIK3CA* while few data are available regarding MSI or MMR IHC tests and intra- and inter-tumoral heterogeneity, and its possible variations during the tumor growth process.

In Lynch syndrome, microsatellite instability is the primum movens of tumor carcinogenesis and all tumor cells should be dMMR/MSI with no heterogeneity. By contrast, in sporadic dMMR/MSI CRC and even if microsatellite instability is considered as an early event, MMR deficiency may emerge late in tumor progression with intra-tumoral or inter-tumoral heterogeneity. Indeed, according to Chapusot et al., among 100 sporadic proximal CRCs, eight present an uncommon MMR IHC pattern with loss of MMR expression (MLH1 and MSH6) restricted to small tumor areas and are MSI. Further analyses on other tumor areas finally established the presence of intra-tumoral heterogeneity [76]. Joost et al. collected 14 CRCs with heterogeneous IHC staining patterns that affected at least one of the MMR proteins, that is, MLH1/PMS2 in three tumors, PMS2 in two tumors, MSH2/MSH6 in 10 tumors (of which two also expressed heterogeneity for MLH1/PMS2), and MSH6 in one tumor. Analysis of these 14 sectioned tumor blocks by molecular biology using the pentaplex panel demonstrated intra-tumoral heterogeneity in three out of 14 tumors [71]. On the one hand, Tachon et al. reported a case of CRC with heterogeneous MLH1/PMS2 staining pattern in primary tumor confirmed by MSI testing, MSS in pMMR areas, and MSI in dMMR areas (intra-tumoral heterogeneity) [64]. By contrast, metastatic lymph nodes were pMMR/MSS. On the other hand, in a cohort of 271 CRC Asian patients, all MSI samples (*n* = 39) were dissected into three regions by tumors and analyzed using Bethesda panel and no intra-tumoral heterogeneity was detected [77].

Although they are rare, due to the prognostic and therapeutic impacts of dMMR and MSI status, detection of these atypical cases should be undertaken cautiously. Since most of these cases present atypical staining at MMR IHC, MSI will help to determine the major tumor subclones. In the case of discordance (dMMR/MSS), it may be useful to test at least two tumor areas of primary tumor or metastases with both techniques in order to identify tumor heterogeneity. Finally, large prospective studies are necessary to determine the rate of intra-tumoral and inter-tumoral heterogeneity of microsatellite instability and their impact on prognosis and treatment efficacy. As of now, there are no data concerning the efficacy of ICI in mCRC with intra- and inter-tumoral heterogeneity of dMMR and MSI status. Moreover, to our knowledge, there are no data concerning the dynamic evolution of MMR and MSI status during tumor progression, especially under treatment selection pressure.

## 6. Other Markers of Microsatellite Instability and Response to Immune Checkpoint Inhibitors

Finally, while MSI/dMMR reflects tumor genomic instability, it is not the only marker of response to ICI. Tumor mutational burden (TMB) has been described as predictive of response to ICI, in the subpopulation of MSI/dMMR mCRC [78], but not exclusively [79]. In dMMR CRC, due to frameshift mutations leading to abnormal truncated protein products, several immunogenic neoantigens are generated and explain the high number of tumor-infiltrating lymphocytes in these tumors [80]. For example, cytotoxic T lymphocytes specific to a neoantigen derived from *TGFbetaRII* frameshift mutation have been identified in dMMR CRC harboring this mutation [81]. Two triggering signals are required to initiate adaptive immune response by T cells: MHC-antigen (major histocompatibility complex) peptide recognition by the T-cell receptor and costimulation via a collection of receptors interacting with related ligands on antigen-presenting cells (APCs), thereby avoiding T-cell anergy [82]. CD28 is an example of costimulatory (positive) molecules and is constitutively expressed on the T-cell surface. It binds to B7.1 (CD80) or B7.2 (CD86), which are expressed on APCs and provides the positive costimulatory signal required for T-cell activation and survival [83]. In contrast, B7 molecules also interact with cytotoxic T-lymphocyte-associated antigen 4 (CTLA-4), which is expressed on T cells, to which it transmits an inhibitory costimulatory signal. In many cancers, especially dMMR/MSI mCRC, ligands of co-inhibitory immune checkpoint receptors are upregulated in a cancer cell or tumor microenvironment, leading to T-cell functional exhaustion and unresponsiveness (a state of anergy), and therefore to loss of tumor growth control (tumor escape). Other well-known immune checkpoints include PD-L1/PD-1, MHC class II and lymphocyte activation gene 3 (LAG-3), and galectin-9 and mucin domain-containing protein-3 (TIM-3). These interactions allow negative feedbacks and, using this rationale, several immune checkpoint inhibitors have been developed for cancer treatment (mAbs blocking co-inhibitory immune checkpoint receptors or their ligands). Indeed, hypermutated dMMR/MSI mCRCs, i.e., with high neoantigens load, have high sensitivity to immune checkpoint inhibitors, which reactivate cytotoxic T cells to kill dMMR/MSI tumors cells.

Alteration in *POLE* gene (DNA polymerase epsilon), observed in ≈0.5% of CRCs [84] can be detected somatically or constitutionally and is also responsible for tumor genomic instability, with an ultramutator phenotype and high TMB [85]. POLE encodes the major catalytic and proofreading subunits of the Polε DNA polymerase enzyme complex. The proofreading (exonuclease) function locates and replaces erroneous bases in the daughter strand through failed complementary pairing with the parental strand. High-fidelity incorporation of bases by POLE, coupled with its exonuclease proofreading function, ensures a low mutation rate. Clinically, patients in the POLE ultra-mutated group have been reported to have high sensitivity to ICI [85]. Recent reports suggest an overlap between dMMR and MSI status and POLE mutation.

A study has evaluated the relationship between MSI/dMMR, TMB, and PD-L1 expression in approximately 2000 tumor samples (1395 CRCs) [86]. About one-third of the MSI cases, all types of cancers combined, had TMB-low and only 26% of the MSI cases had positive PD-L1 status. All in all, only 0.6% of cancers combined the three positive markers (MSI, TMB high, and PD-L1 positive), although overlaps varied according to tumor type. Only 1.3% of CRCs combined the three positive markers (MSI, TMB high, and PD-L1 positive) but 5.7% were MSI, 6.7% TMB high, and 7.2% PD-L1 positive. In this CRC population, 5.4% of overlap MSI and TMB high and 1.6% of overlap MSI and PDL1 positive were observed [86]. For the moment, PD-L1 expression is not a proven biomarker to select mCRC patients for treatment with ICI [87] and TMB determination is too time-consuming and expensive for routine clinical practice. In spite of not being perfect, dMMR/MSI remains an established marker of response to ICI in mCRC and its determination is crucial [12]. Nevertheless, in mCRCs with discordances between IHC of MMR proteins and MSI, molecular testing determination of TMB could help to select mCRC patients for treatment with ICI. New biomarkers to select mCRC patients eligible for ICI are under investigation, especially immunoscore, which is being prospectively evaluated by our group in the Pochi trial (xelox, bevacizumab plus pembrolizumab in pMMR/MSS mCRC with high immunoscore).

Finally, molecular alterations associated with dMMR/MSI tumors, other than impairment of the four core MMR proteins, have recently been described. These alterations mostly impact histone methyltransferase or demethylase, with examples such as depletion of chromatin regulator SETD2 (set domain containing 2, methyltransferase) or deletion of FANCJ (Fanconi anemia, helicase) or overexpression of KDM4 (H3K36me2/me3 demethylase), which reduces the abundance of MSH6 [88,89,90]. In the Awwad et al. study, KMD4 overexpression led to disruption of MSH6 foci formation during S phase by demethylating its binding site, and resulted in a DNA mismatch repair system [90]. Genes involved in histone modification, such as SETD2, were significantly more mutated in older patients (≥65 years) and this enzyme modifies histone proteins associated with DNA that control the regulation of gene expression and DNA replication and prevent the association of MMR proteins with damaged DNA, thereby preventing DNA mismatch repair [91].

## 7. Perspectives of MMR Immunohistochemistry and DNA Microsatellites Testing

Aside from multiple tumor area testing and systematic double screening by MMR IHC and MSI tests, there are other ways to avoid false positives or false negatives, such as exploration of other microsatellite markers.

### 7.1. The Role of HSP110 Protein in the Diagnosis of Microsatellite Status

HSP110, a chaperon protein with a T_17_ mononucleotide repeat located within intron 8, described in CRC in 2011, shows a remarkably monomorphic profile in non-tumor tissue and is an interesting candidate for microsatellite instability assessment with IHC and in molecular biology (Figure 7 and Figure 8) [69,92]. Buhard et al. have suggested that *HSP110 T_17_* deletion is present in all true dMMR/MSI cases and should be used as a complementary test in discordant cases [69]. Indeed, from 70 patients with CRC considered to be at high risk of LS, 46 displayed unambiguous MSS status with the pentaplex panel and no aberrant HSP110 was detected in 45 of these 46 tumors (98%). For one patient with aberrant HSP110, MSI status was confirmed by IHC showing loss of MSH2 expression in the tumor. By contrast, in the Kim et al. study, 12% (*n* = 20/168) of MSI CRCs were not associated with instability of *HSP110 T_17_* [93], however, based on our experience, establishment of HSP110 status by IHC or molecular technique (*HSP110 T_17_* deletion) does not show 100% correlation and molecular determination of *HSP110 T_17_* deletion can be difficult. Finally, while more data are needed before using *HSP110 T_17_* in routine clinical practice for MSI status determination, *HSP110 T_17_* could be of help in difficult cases.

### 7.2. A Larger Panel of Microsatellites for Better Detection of Instability

To improve MSI detection, another way is to analyze a large panel of microsatellites instead of only five (i.e., pentaplex or Bethesda panels). Next-generation DNA sequencing (NGS), now routinely used for determination of molecular alterations in most cancers, can allow comprehensive investigation of multiple microsatellite loci simultaneously, while using appropriate computational tools. Several tools have been developed and compared to standard procedures [70,94,95]. The mSing method incorporating from 15 to 2957 microsatellite markers, has demonstrated 96.4% to 100% sensitivity and 97.2% to 100% specificity as compared to the pentaplex panel [94]. MSIsensor and MANTIS (microsatellite analysis for normal tumor instability), other computational tools for MSI detection with NGS, have also demonstrated their feasibility, although requiring non-tumor DNA [95,96]. MSIsensors scan a reference genome to locate homopolymers and microsatellites and then record homopolymers of at least 5 bp length and microsatellites of maximum repeat unit length five from the reference genome and each site is saved in a loci file for subsequent analysis. MANTIS uses a set of mono- to penta-nucleotide repeat microsatellites to detect MSI, by individually computing and aggregating the differences between the allele length distribution of each locus of matched tumor and normal samples to achieve an average distance score (zero, fully stable and two, fully unstable); a score threshold of 0.4 is recommended to diagnose MSI in tumors. MANTIS has displayed superior performance compared to the other tools (MANTIS, MSISensor, and mSINGS), having the highest overall sensitivity and specificity, even with loci panels of varying size. Nevertheless, interest in classifying cases with discordant MMR IHC and MSI results has never been explored. Moreover, NGS experiments are time-consuming and more expensive than standard MMR IHC/MSI techniques, notably due to the complex bioinformatic analysis required. The benefit and the place of this technique in this indication, therefore, remain to be precisely determined [97].

### 7.3. Tumor Circulating DNA to Overcome Tumor Heterogeneity

As previously highlighted, correct identification of dMMR and MSI status can be impaired by tumor heterogeneity, samples with few tumor cells or discordance between the two tests. Analysis of circulating tumor DNA (ctDNA) could be another method likely to easily determine MSI status. Tumor DNA is released into the bloodstream by exosome, secretion or necrosis and apoptosis. ctDNA allows direct analysis at a given time of all the molecular alterations present in the tumor and the metastases, in a minimally invasive way (blood test). To our knowledge, only two studies have reported the detection of MSI status using ctDNA [98,99]. In the Deng et al. study, 13 MSI CRCs based on standard MSI analysis (six mononucleotides: *NR-27, NR-21, BAT-26, BAT-25, NR-24*, and *MONO-27*), were correctly classified “MSI” using the NGS technique on tumoral tissue (analyses of the same microsatellite loci). On ctDNA of the same 13 patients, the standard MSI technique (by PCR) failed to detect MSI status while the NGS technique on ctDNA (the same as the one used for the tissue) correctly identified all MSI cases [98]. In the second study with pan-cancer, ctDNA testing using the Guardant360 (Guardant Health Clinical Laboratory, Redwood City, CA, US) NGS kit incorporating 99 putative microsatellite loci, accurately detected 87% (71/82) of tissue MSI-H and 99.5% of tissue microsatellite stability (863/867) with an overall accuracy of 98.4% (934/949) [99].

ctDNA testing allows non-invasive MSI testing at diagnosis and also provides an opportunity to follow the kinetics of MSI status. Monitoring MSI status at several time-points of tumor progression, especially at treatment start and at relapse or progression, may be of major interest for monitoring response to ICI (i.e., selection of pMMR/MSS subclones?) [100]. ctDNA testing could also help with regard to the tumor heterogeneity issue as it reflects all the tumor subclones at a specific time [101]. Multiple samplings would not be necessary as a single blood sample could detect all major clones located in the primary tumor and metastases as well. Nevertheless, as of now, we do not know whether dynamic change of MSI during carcinogenesis exists. Finally, as MMR IHC and MSI discordances are sometimes caused by insufficient sample quality or tumor cell quantity, ctDNA testing could also be an interesting substitute [102]. Nevertheless, more data are needed to validate MSI detection by ctDNA before application in routine clinical practice, which is ongoing in trials using ICI in dMMR/MSI tumors [98].

New techniques for MSI assessment will probably help to more accurately diagnose dMMR/MSI CRC and classify discordant cases such as MSI or MMS. However, while increasing the number of markers to define the MSI status, prognosis of the technique may not significantly improve. On the contrary, false positives may arise. Detection of cases with a low level of instability may be useful but can also be without a relevant impact on routine clinical practice. Indeed, one of the urgent questions to be addressed would be whether discordant cases or those with a low level of microsatellite instability could benefit from immune checkpoint inhibitor.

## 8. Conclusions

MMR/MSI screening is crucial not only for LS screening but also in therapeutic management, especially since the ICI revolution [89]. Indeed, patients with dMMR/MSI tumor have drawn a major benefit from immunotherapy, whatever the tumor type, with a rate of disease control reaching 80% and overall survival superior to three years in chemoresistant mCRC cases [10,12,103]. Consequently, all mCRCs should be tested for MMR and MSI status.

Since ICI is ineffective in pMMR/MSS CRC, as well as expensive and liable to induce severe side effects, it is essential to avoid false positive dMMR/MSI and false negative pMMR/MSS, and also to classify discordant cases (pMMR/MSI and dMMR/MSS) [104]. As recommended in recent trials with ICI, screening for ICI treatment in mCRC can be done by means of one test, MSI or MMR IHC, but in the case of a positive test or difficulties interpreting the results, the second test should be performed before treatment with ICI to confirm concordant dMMR/MSI results [67,68].

All the pitfalls of the different techniques must be known and any discordances should be investigated, in order to achieve the following: 1-control MMR IHC and MSI results, 2-check sample quality, 3-perform new tests on multiple tumor areas or sites, 4-test other microsatellites or ctDNA if techniques are available (or else, send sample to an expert team), and 5-perform MMR germline testing in the case of suspicion of LS. Indeed, rates of “true” discordant cases remain low. New techniques for MSI assessment (NGS, ctDNA, etc.) will probably help to more accurately diagnose dMMR/MSI CRC, which is a major challenge for these patients to have access to immunotherapy.

## Figures and Tables

**Figure 1 cancers-11-01567-f001:**
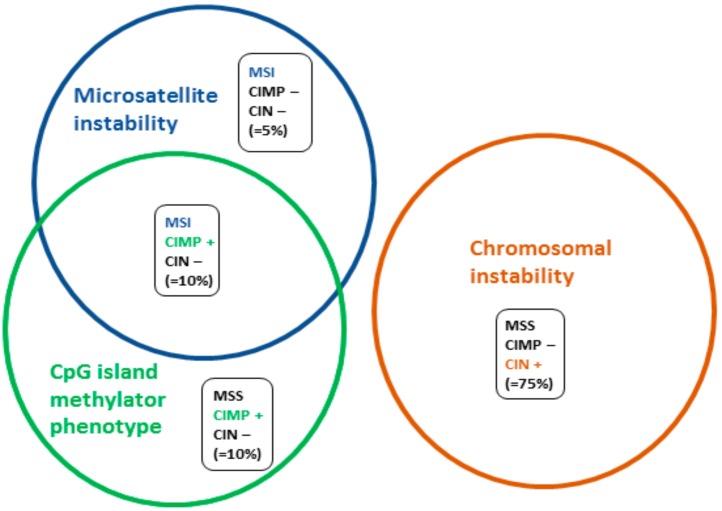
Simplified molecular subgroups of colorectal cancers. There are three major mechanisms of colorectal carcinogenesis, 75% of chromosomal instability, 20% of DNA methylation, and 15% of microsatellite instability or deficient DNA mismatch repair [1]. MSS: microsatellite stability, MSI: microsatellite instability, CIMP: CpG island methylator phenotype, CIN: chromosomal instability.

**Figure 2 cancers-11-01567-f002:**
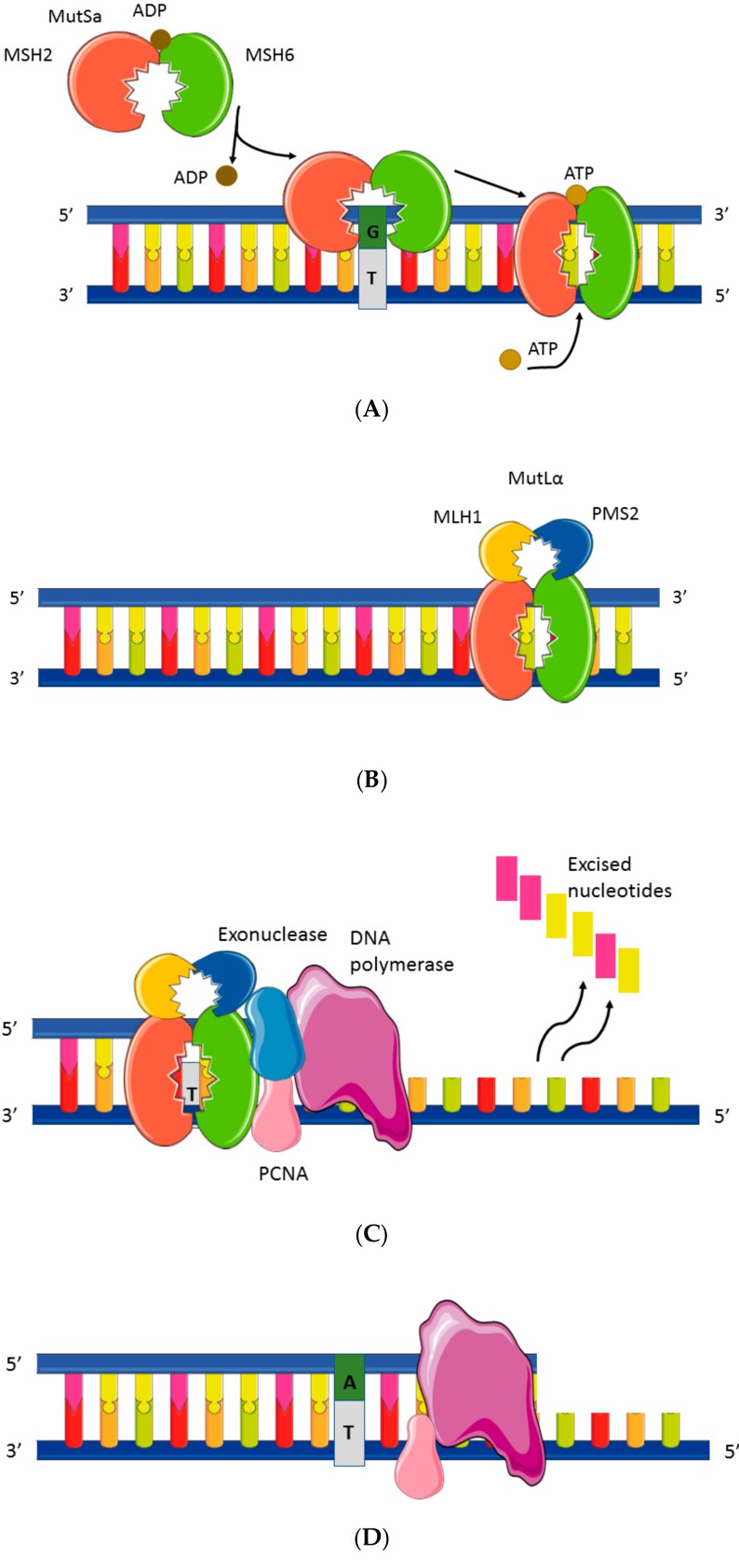
Mismatch repair mechanism. (**A**) Single mismatch, (**B**) DNA MMR protein sliding clamp, (**C**) exonuclease complex, and (**D**) resynthesis. The complex MutSa recognizes single base pair mismatch and surrounds the DNA like a clamp and then the MutL complex comes and links to MutSa. Different enzymes (PCNA and DNA polymerase) then intervene to excise the errors and to resynthesize the DNA. PCNA: proliferating cell nuclear antigen, ADP: adenosine diphosphate, ATP: adenosine triphosphate, and MMR: mismatch repair. Proteins from the DNA repair system: MSH2, MSH6, PMS2, and MLH1.

**Figure 3 cancers-11-01567-f003:**
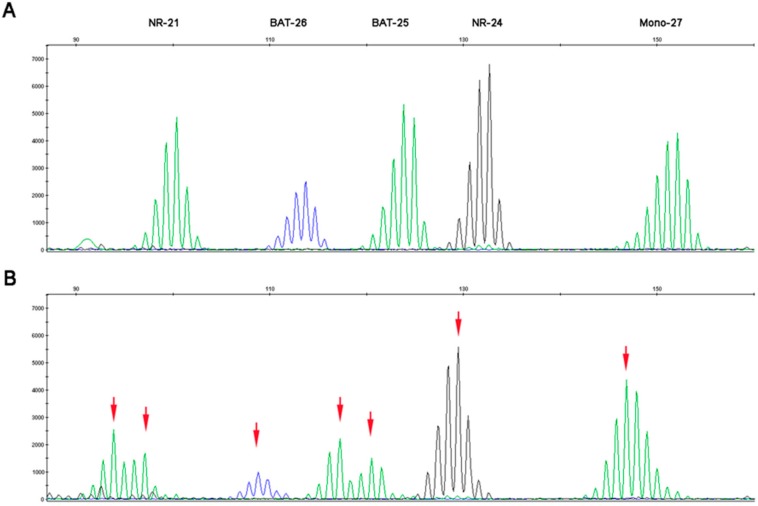
MSS and MSI profiles using the pentaplex panel. (**A**) MSS profile of the five consensus mononucleotide repeats and (**B**) MSI profile with 5 unstable mononucleotide repeats. Red arrows indicate microsatellite instability.

**Figure 4 cancers-11-01567-f004:**
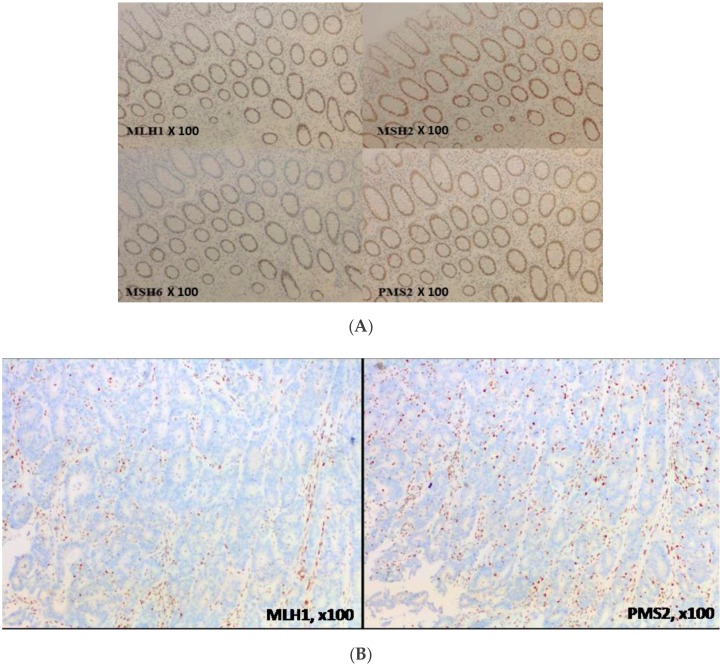
Immunochemistry of MMR proteins. (**A**) MMR-proficient (pMMR) tumor, normal colonic mucosa, no loss of expression of MMR proteins and (**B**) deficient MMR (dMMR) tumor with loss of MLH1 and PMS2 expression.

**Figure 5 cancers-11-01567-f005:**
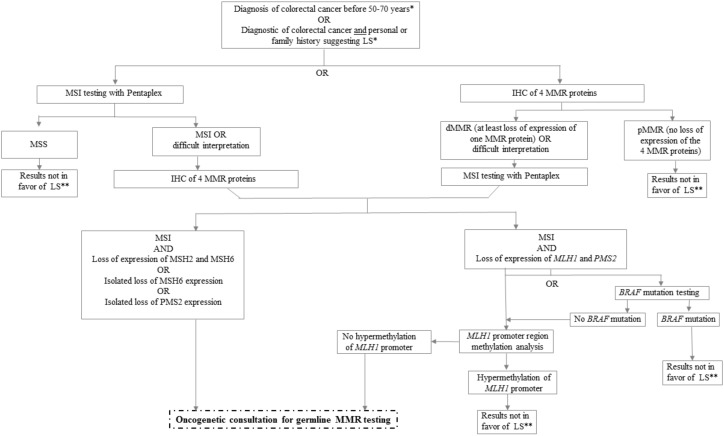
dMMR/MSI screening of colorectal cancer (CRC) for suspicion of Lynch syndrome. * According to the country and the guidelines, universal testing of CRC is recommended only if revised Bethesda criteria are met. ** A result that is not in favor of diagnosis of Lynch syndrome must be interpreted according to the patient’s family history and if Lynch syndrome or another genetic predisposition is suspected, the patients must be referred to oncogenetic consultation. dMMR: deficient mismatch repair, IHC: immunohistochemistry, LS: Lynch syndrome, MSS: microsatellite stability, MSI: microsatellite instability, and pMMR: proficient mismatch repair.

**Figure 6 cancers-11-01567-f006:**
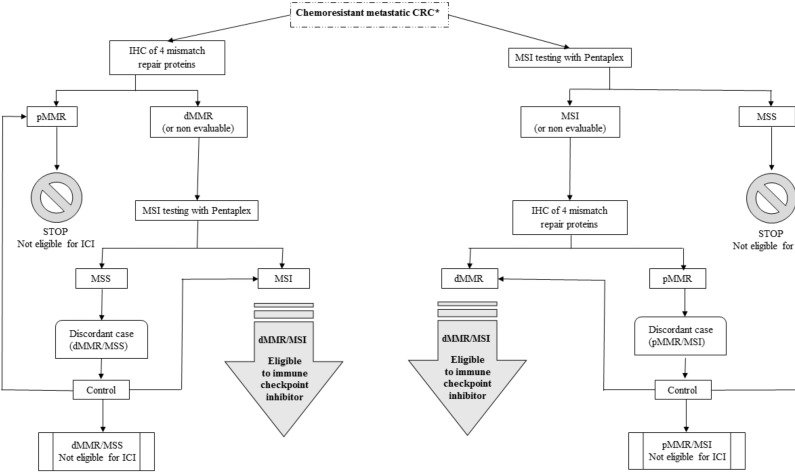
dMMR/MSI screening of metastatic CRC for eligibility to immune checkpoint inhibitors. * If no suspicion of Lynch syndrome. CRC: colorectal cancer, dMMR: deficient mismatch repair, IHC: immunohistochemistry, LS: Lynch syndrome, MSS: microsatellite stability, MSI: microsatellite instability, and pMMR: proficient mismatch repair.

**Figure 7 cancers-11-01567-f007:**
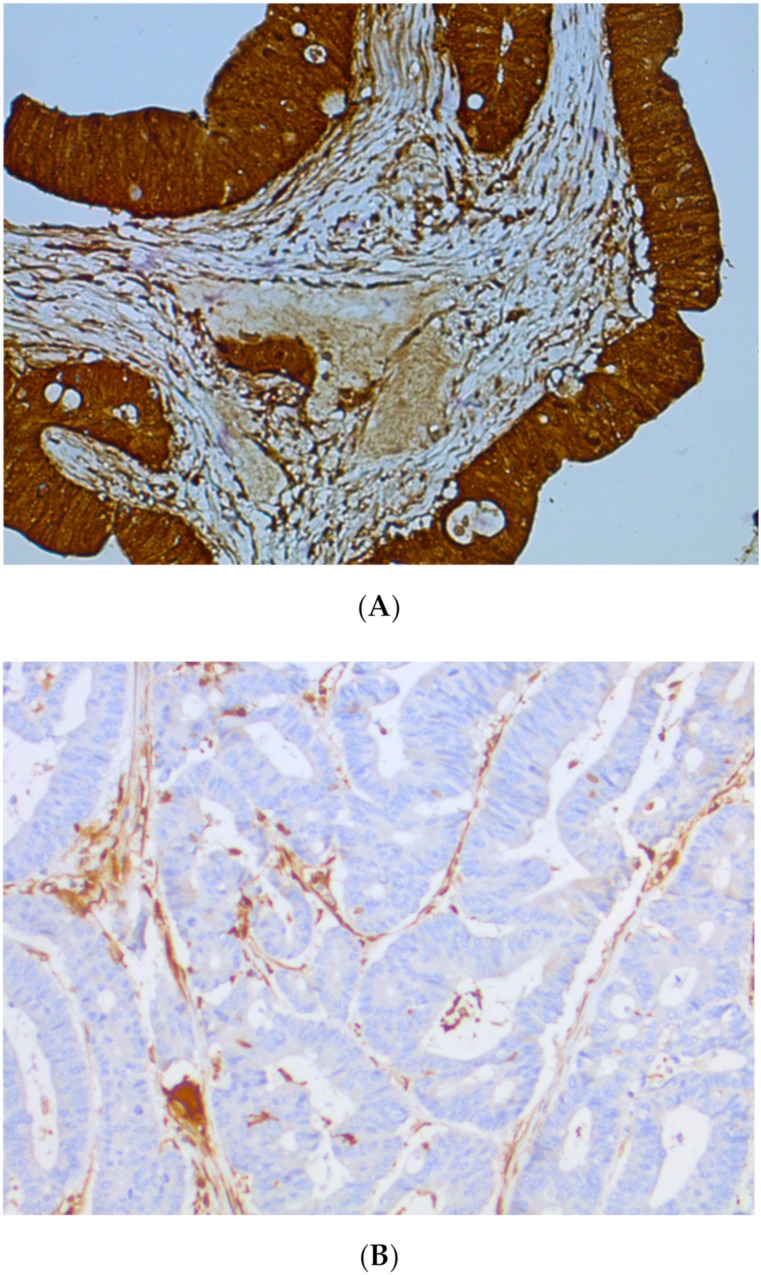
Immunohistochemistry of HSP110. (**A**) HSP110 expressed, ×20 and (**B**) loss of expression of HSP110, ×20.

**Figure 8 cancers-11-01567-f008:**
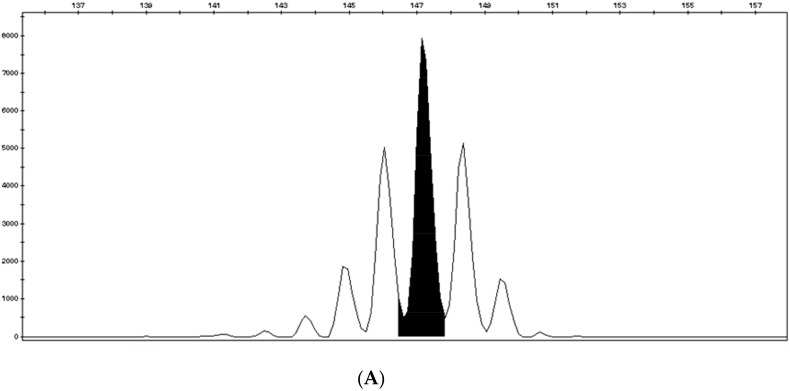
Microsatellite instability testing of *HSP110 T17*. (**A**) MSS tumor with T16/T16 phenotype (major peak at 147 bp) and (**B**) MSI tumor with large deletion of *HSP110 T17,* patient with a T16/T17 phenotype (peaks at 147 and 148 bps in the polymorphic zone). The major peak selected to determine size of the deletion is the peak at 147 bps and the black arrow corresponds to a deletion of 4 bps.

**Table 1 cancers-11-01567-t001:** Revised Bethesda criteria [14].

Colorectal cancer diagnosed in a patient less than 50 years of age Presence of synchronous, metachronous colorectal, or other HNPCC-associated tumors *, regardless of age Colorectal cancer with the MSI histology ^†^ diagnosed in a patient less than 60 years of age ^§^ Colorectal cancer diagnosed in one or more first-degree relatives with an HNPCC-related tumor, with one of the cancers being diagnosed under age 50 years Colorectal cancer diagnosed in two or more first- or second-degree relatives with HNPCC-related tumors, regardless of age

*: Hereditary nonpolyposis colorectal cancer (HNPCC)-related tumors include colorectal, endometrial, stomach, ovarian, pancreas, ureter and renal pelvis, biliary tract, and brain (usually glioblastoma as seen in Turcot syndrome) tumors, sebaceous gland adenomas and keratoacanthomas in Muir–Torre syndrome, and carcinoma of the small bowel. ^†^: Presence of tumor-infiltrating lymphocytes, Crohn’s-like lymphocytic reaction, mucinous/signet-ring differentiation, or medullary growth pattern. ^§^: There was no consensus among the workshop participants on whether to include the age criteria in guideline three above; participants voted to keep less than 60 years of age in the guidelines.

**Table 2 cancers-11-01567-t002:** The mutation frequencies of mismatch repair gene in Lynch syndrome [37,38,39].

Mismatch Repair Gene	Mutation Frequency
*MSH2*	50%
*MLH1*	30–40%
*MSH6*	7–10%
*PMS2*	<5%
*EPCAM*	1–3%
Constitutional *MLH1* epimutation	1–3%

**Table 3 cancers-11-01567-t003:** Rate of discordance between MMR immunohistochemistry and MSI testing in CRC.

Series *	Number of Patients	Population	MMR IHC	Molecular MSI Testing	Discordance Rates
Lindor NM et al., 2002 [54]	1144	From multiple centers from the Cooperative Family Registry for Colon Cancer Studies: USA, Australia, and Canada	2 proteins (MLH1 and MSH2)	10 markers: *BAT25, BAT26, BAT40, BAT34C4, D5S346, D17S250, ACTC, D18S55, D10S197,* and *MYCL.* or 6 markers: *D5S346, TP53, D18S34, D18S49, D18S61, ACTC* and *BAT 26*	2.4%
Hatch et al., 2005 [55]	262	CRC with complete resection	4 proteins (MLH1, MSH2, MSH6 and PMS2)	NCI panel (*D5S346, BAT25, BAT26, D2S123*, and *D17S250*)	5.4%
Pinol et al., 2005 [56]	1222	CRC in Spain	2 proteins (MSH2 and MLH1)	*BAT26* ± *BAT-25, D5S346, D2S123*, and *D17S250*	2.8%
Watson et al., 2007 [57]	Cohort 1: 68 Cohort 2: 208	CRC patients younger than 60 years (*BRAF* mutated CRC are excluded in cohort 1)	4 proteins (MLH1, MSH2, MSH6 and PMS2)	Single microsatellite: *BAT26*	Cohort 1: 1.4% Cohort 2: 1%
Yuan L et al., 2015 [58]	296	CRC patients fulfilled revised Bethesda criteria	4 proteins (MLH1, MSH2, MSH6 and PMS2)	Bethesda panel	1%
Chen et al., 2018 [59]	569	Chinese monocentric study with only CRC	4 proteins (MLH1, MSH2, MSH6 and PMS2)	Bethesda panel	8.1%
Cohen et al., abstract ESMO 2018 [60]	92	CRC only	4 proteins (MLH1, MSH2, MSH6 and PMS2)	Pentaplex panel	9.1%
Jaffrelot M et al., abstract JFHOD 2019 [61]	2528	Patients with dMMR tumors (CRC, endometrium, non-colorectal digestive cancers and others)	4 proteins (MLH1, MSH2, MSH6 and PMS2)	Pentaplex panel	1.1%

CRC: colorectal cancer, IHC: immunohistochemistry, MMR: mismatch repair, MSI: microsatellite instability, and NCI: national institute cancer. * Only studies with more than 50 patients were included in the Table.

**Table 4 cancers-11-01567-t004:** Main causes of discordances and quality criteria to prevent them.

Causes of Discordance	Quality Criteria to Prevent Discordance	
	MMR IHC	Molecular DNA testing
Low tumor cells [62]	Selection of a specific area with the highest rate of tumor cells	Macrodissection or selection of tumor sections enriched in tumor cells (≥20%)
Pre-analytical difficulties [28]	Use formol 4% (not Bouin’s fixative), protocol standardization efforts, participation in national and international quality assessment	Protocol standardization efforts, participation in national and international quality assessment
Non-expert physician [29]	Participation in training sessions and request for rereading by expert if necessary	
Neoadjuvant treatment [63]	Testing on pretherapeutic samples	
Polymorphisms in non-Caucasian ethnic groups	-	Testing of paired tumor and non-tumor tissues
Discordance of tumor biopsy	Testing of the complete surgical resection	
Heterogeneous IHC pattern (Tumor heterogeneity suspected) [64]	Multiple sampling	-

MMR: mismatch repair and IHC: immunohistochemistry.

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
