# Peer review of "Microsatellite Instability: Diagnosis, Heterogeneity, Discordance, and Clinical Impact in Colorectal Cancer"

_cancers, 2019, doi:10.3390/cancers11101567_

Round 1

Reviewer 1 Report

This review talks about the importance of MSI status within colorectal cancer tumors in the decision-making for treatment and/or diagnosis, specially focusing on the difficulty for the techniques carried out for this aim. In my opinion, presented as a review, lot of points should be revised, and maybe modified. I underline some points that should be changed or at least better explained.  

In Figure 1 the overlapping of the three main colorectal carcinogenetic pathways are is more complex than the way shown by the authors (see Snover DC et al. Hum Pathol 2010).  Gene expression should be written in italics (e.g. MSH2...).  The indications of chemotherapy for stage II CRCs is not clear explained (page 4).  In the Introduction, and also in some points within the text, there is a lack of explanation about the use immune checkpoint inhibitors for MSI carriers (at least shortly it would be fine to explain it).  And finally, I think there is an important problem about the use of the references along the whole text (e.g.):Some of them seem to be missing: line 104-107, page 4...where are the 75% and 25% from? The same for the 20% in line 312, page 10; or in lines 526-537, where there is a large text without any...In line 216, page 8, authors tell about the actual standard panel...with a reference of 2002 (16).  I would suggest to change the term "constitutional dMMR/MSI tumors", and maybe using germline dMMR, in order to avoid confusion with constitutional MMR deficiency syndrome (biallelic germline MMR deficiency). And why the authors use "reliable" for useful technique talking about ctDNA analysis for MSI status, when there are only two studies published until today? Minor changes: Add "respectively" in the expression of line 181, page 7. 

Author Response

Reviewer 1

This review talks about the importance of MSI status within colorectal cancer tumors in the decision-making for treatment and/or diagnosis, specially focusing on the difficulty for the techniques carried out for this aim. In my opinion, presented as a review, lot of points should be revised, and maybe modified. I underline some points that should be changed or at least better explained. 

Specific Comments

In Figure 1 the overlapping of the three main colorectal carcinogenetic pathways are is more complex than the way shown by the authors (see Snover DC et al. Hum Pathol 2010). 

We agree with the reviewer that the overlap is more complex than presented in Figure 1. Since the review is focused on dMMR/MSI, we have not detailed this point. We have now mentioned it in the manuscript (Figure 1, page 2) and have added some references to briefly describe the rare overlap between CIMP and CIN and CIN and MSI phenotypes (page 1).

Gene expression should be written in italics (e.g. MSH2...). 

All of the modifications have been done in the manuscript accordingly: MSH2 when we refer to the protein and MSH2 in italics when we refer to the gene.

The indications of chemotherapy for stage II CRCs is not clear explained (page 4),

We agree with the reviewer’s comment and consequently we have changed the sentence and have added some clarifications (page 4): “Despite better prognosis of dMMR//MSI phenotype in stage II-III CRCs, tumors do not benefit from adjuvant fluoropyrimidine chemotherapy (Ribic et al. 2003). Consequently, in view of good dMMR/MSI prognosis and chemoresistance to fluoropyrimidine, adjuvant chemotherapy for stage II CRCs is not recommended for low-risk or high-risk cancers. Nevertheless, for high-risk stage II dMMR/MSI CRCs with criteria of very poor prognosis like T4 stage and/or vascular emboli, oxaliplatin-based adjuvant chemotherapy remains to be discussed case by case (Tougeron et al. 2016).

In the Introduction, and also in some points within the text, there is a lack of explanation about the use immune checkpoint inhibitors for MSI carriers (at least shortly it would be fine to explain it) :

We agree that more explanations are needed; therefore we have added details of immune checkpoint inhibitors and mechanism page 14-15.

And finally, I think there is an important problem about the use of the references along the whole text (e.g.):Some of them seem to be missing: line 104-107, page 4...where are the 75% and 25% from? The same for the 20% in line 312, page 10; or in lines 526-537, where there is a large text without any... In line 216, page 8, authors tell about the actual standard panel...with a reference of 2002 (16). 

We agree that references can be scattered throughout the manuscript. It was mostly to avoid repetition of references in the same paragraph as they were cited elsewhere in the manuscript, just before or after. Nevertheless, we have taken into consideration the reviewer’s comment and have added references at the indicated location as well as in parts of the manuscript where references may be missing. Regarding the reference, initially numbered 16, it is the princeps reference about the pentaplex method, method still used worldwide to determine the MSI status in CRC, therefore, we think it is important to cite this reference in the manuscript. Nevertheless, we have added in the paragraph another reference that comparing the pentaplex panel with the Bethesda panel (Murphy et al. 2006).

I would suggest to change the term "constitutional dMMR/MSI tumors", and maybe using germline dMMR, in order to avoid confusion with constitutional MMR deficiency syndrome (biallelic germline MMR deficiency).

We agree that the term “constitutional” could be confusing between Lynch syndrome versus CMMRD.  All the modifications have been made in the manuscript (see page 1,8 and 10).

And why the authors use "reliable" for useful technique talking about ctDNA analysis for MSI status, when there are only two studies published until today?

We have removed this term from the manuscript.

Minor changes: Add "respectively" in the expression of line 181, page 7. 

The modification has been made in the manuscript.

Reviewer 2 Report

Dear Authors, 

Minor

Figures

Minor

Fig 2.A MutS should be MutSa

Fig 3 be consistent name the figures Fig 3.A and Fig 3

Fig 4.A change Fug 4.A to Fig 4.A

Major

The manuscript addresses the four core proteins causing MSI, the authors neglected to address other causes that address MSI caused by indirectly. For example depletion of chromatin regulator SETD2 or overexpression of KDM4, (an H3K36me2/me3 demethylase) reduce the abundance of MSH6 leading to increase of MSI. In addition it is also reported that deletion of FANCJ is been associated in to increase MSI.

Author Response

Reviewer 2

Minor

Fig 2.A MutS should be MutSa Fig 3 be consistent name the figures Fig 3.A and Fig 3 Fig 4.A change Fug 4.A to Fig 4.A

 All the changes highlighted by the reviewer 2 in the figures have been made.

Major

The manuscript addresses the four core proteins causing MSI, the authors neglected to address other causes that address MSI caused by indirectly. For example depletion of chromatin regulator SETD2 or overexpression of KDM4, (an H3K36me2/me3 demethylase) reduce the abundance of MSH6 leading to increase of MSI. In addition it is also reported that deletion of FANCJ is been associated in to increase MSI.

We agree with the reviewer that other mechanisms can induce MSI tumors. Since the review is focused on heterogeneity and discordance of dMMR/MSI status, we did not detail these points. Nevertheless, we have now mentioned them in the manuscript (page 15-16) and have added some references.

Reviewer 3 Report

This is a nice overview of current challenges in the determination of the MSI status. Overall, I think the paper is a bit too long and could be shortened. The authors may be able to shorten some parts. In addition to that, I have the following suggestions:

line 75-76: provide reference for MSI pathway proportions.
line 187: would not say that 5-15% percent is frequent, it is rather not rare.
line 239: LS not SL.
line 241: provide reference for the percentages reported.
line 342-57: Redundant. Better add references to table 4 and omit the text in lines 342-357.
line 554-56: If more markers are used to determine MSI status this does not necessarily mean that the prediction of ICI treatment response or prognosis is improved. Some of the existing/new/minor MMR deficiencies could be meaningless (not clinically relevant). Maybe this is worth a discussion.

Author Response

Reviewer 3

This is a nice overview of current challenges in the determination of the MSI status. Overall, I think the paper is a bit too long and could be shortened. The authors may be able to shorten some parts.

Indeed, our review is about a broad subject but we think every paragraph is necessary and gives interesting and different information. Besides, other reviewers have asked us to develop certain points even more. Nevertheless, some parts of the manuscript have been shortened (page 12 for example).

In addition to that, I have the following suggestions:

line 75-76: provide reference for MSI pathway proportions.

We have added a reference to justify the proportion of each carcinogenetic pathways (page 4).

line 187: would not say that 5-15% percent is frequent, it is rather not rare.

We agree and have made the modification (page 7).

line 239: LS not SL.

The typing error has been corrected (page 8)

line 241: provide reference for the percentages reported.

The references have been added in the manuscript (page 8).

line 342-57: Redundant. Better add references to table 4 and omit the text in lines 342-357.

We agree with the reviewer’s suggestion. We have now deleted the text in the revised manuscript and it helps to shorten a bit our manuscript (page 12-13).

line 554-56: If more markers are used to determine MSI status this does not necessarily mean that the prediction of ICI treatment response or prognosis is improved. Some of the existing/new/minor MMR deficiencies could be meaningless (not clinically relevant). Maybe this is worth a discussion.

Thank for this valuable comment. This subject is now discussed page 19. We agree that while new techniques can detect low rate of microsatellite instability, it does not necessarily correlate with ICI treatment response or prognosis.

Round 2

Reviewer 1 Report

Review the English editing again. The other changes that have been requested, have been correctly responded. 

Author Response

Point-by-point response to reviewers’ comments

All the new modifications have been performed using “Track Changes” function in Microsoft Word.

Reviewer 1

Review the English editing again. The other changes that have been requested, have been correctly responded.

Jeffrey Arsham, an American medical translator, has reviewed English-language.
